# Sleepiness and Fatigue as Consequences of Cumulative Sleep Restriction: Insights from Fine-Grained Subjective Measures and Skin Temperature in the Field

**DOI:** 10.3390/clockssleep7030051

**Published:** 2025-09-19

**Authors:** Vaida T. R. Verhoef, Karin C. H. J. Smolders, Geert Peeters, Sebastiaan Overeem, Yvonne A. W. de Kort

**Affiliations:** 1Human-Technology Interaction, Department of Industrial Engineering and Innovation Sciences, Eindhoven University of Technology, P.O. Box 513, 5600 MB Eindhoven, The Netherlands; 2Center for Sleep Medicine Kempenhaeghe, 5591 VE Heeze, The Netherlands; 3Department of Electrical Engineering, Eindhoven University of Technology, 5612 AP Eindhoven, The Netherlands

**Keywords:** sleep restriction, time-of-day variation, sleepiness, fatigue, measurement convergence, skin temperature

## Abstract

Diagnosis and monitoring of daytime sleepiness remain challenging and are strongly reliant on subjective assessments. To revisit common monitoring tools and explore new assessment modalities, we investigated the response of daily and momentary subjective measures of sleepiness and fatigue and skin temperature to sleep restriction, assessed between- and within-day variations in these responses, and studied their convergence. Seventeen healthy participants (aged 19–32 years, seven females, ten males) participated in a field study employing ecological momentary assessment. After a one-week baseline, two sleep conditions (4 h/night vs. 7–9 h/night, actigraphy-controlled; three nights each) were counterbalanced across participants. During the experimental conditions, sleepiness and fatigue were assessed using subjective rating scales administered in daily diary questionnaires and experience sampling questionnaires (10 notifications per day), while distal and proximal skin temperatures were continuously recorded. Results revealed significant effects of sleep restriction on distal and proximal skin temperature and daily reports of sleepiness and fatigue, independent of the number of sleep-restricted nights. The effects on momentary assessments were moderated by day, reflecting a cumulative effect of the sleep restriction from Days 1 to 3. The effects of sleep restriction on momentary sleepiness and fatigue ratings or hourly skin temperature metrics were not significantly moderated by time of day. Hourly skin temperatures (distal skin temperature and temperature gradient) were significantly related to subjective sleepiness and fatigue. In conclusion, all self-reports were sensitive to the sleep restriction, but momentary assessments illustrate the manipulation’s cumulative effects and captured temporal dynamics in sleepiness and fatigue within days. This investigation showed overlap between sleepiness and fatigue experiences as reflected in medium to strong associations. Skin temperature correlates with momentary subjective sleepiness (and fatigue); however, considering it a proxy for daytime sleepiness remains exploratory.

## 1. Introduction

Daytime sleepiness remains a challenging symptom to detect and treat [1]. This challenge may arise partly due to the different perspectives among scientists concerning the definition of daytime sleepiness. For some, it is strictly the obtrusive tendency to fall asleep during the day [2]. Still, for others, daytime sleepiness is a more complex construct that groups related symptoms in three dimensions: proneness to fall asleep during the day, continuous feelings of drowsiness or lack of alertness, and automatic behaviors [3,4,5]. To complicate matters, terms related to sleepiness and fatigue are often used interchangeably by patients and these constructs have been found to overlap [6]. While both subjective sleepiness and fatigue have been hypothesized by Hossain and colleagues to independently coexist as manifestations of sleep disorders [7], assessments of fatigue are sometimes performed with items referring to sleepiness [8], further blurring the lines. Moreover, subjective scales of fatigue remain largely un-validated in clinical populations affected by sleep disorders [9].

In addition to different definitions, multiple metrics and assessments are available: some physiological, some behavioral, and some subjective [1,10,11,12,13]. Notwithstanding the variety of measures, self-reports—defined as “starting points in the assessment of Excessive Daytime Sleepiness (EDS)” [14]—currently represent the most common tools used to monitor sleepiness [5]. The Epworth Sleepiness Scale (ESS) remains a pillar for the diagnosis of EDS, often used as a second step after an in-depth assessment of the patient history. The scale typically instructs participants to rate the likelihood of falling asleep in eight common situations over the last two weeks. Research has shown that increased levels of sleepiness rated with the ESS match behavioral [10,15] and electrophysiological assessments. Although researchers are actively considering these behavioral and physiological metrics as potential additions to the diagnosis of EDS [16], most of these are currently not validated amongst the various clinical populations affected by EDS [10]. The difficulty in assessing daytime sleepiness is also shown by an increasing number of studies raising questions on the reliability of the ESS over time and within persons. A systematic review on the ESS highlighted a lack of internal consistency and weak correlations with other measures of daytime sleepiness [17], an observation reiterated by a subsequent review [10]. Additionally, machine learning analyses [18] conducted on a large cohort of data revealed that only a small amount of variance in ESS scores (7.15–10.0%) could be explained by the set of predictors including sleep-related and medical features. Moreover, the authors uncovered a low correlation between the ESS and other daytime sleepiness proxies (feeling unrested, number of naps, etc.). Taken together, these observations push us to question the (single) use of the ESS in the diagnosis or assessment of EDS [13,18]. Moreover, if we consider one of the multidimensional definitions of EDS [3], the ESS only covers one of the dimensions: the propensity to fall asleep.

The Karolinska Sleepiness Scale (KSS) and the Stanford Sleepiness Scale (SSS) seem to primarily address the second dimension of EDS as defined by Lopez et al. [3], by assessing, respectively, feelings of sleepiness or experiences related to drowsiness or a lack of alertness. Nevertheless, the scales do assess sleepiness on a momentary basis and hence do not necessarily capture the continuity of the symptom over a long period of time, as Lopez et al. suggest [3]. Yet, they offer the possibility to assess sleepiness repetitively throughout the day and thus explore stability or variations in momentary sleepiness states within and between days. In fact, prior studies have highlighted time-of-day-dependent variations in levels of sleepiness [19,20], also as a function of sleep duration [21]. Unfortunately, most studies exploring the temporal dynamics in sleepiness have been performed under controlled conditions in laboratories, requiring validation in real-life situations.

While it is important to revisit self-reports, it is also essential to investigate reliable physiological proxies of sleepiness or fatigue further. Metrics derived from electrophysiological signals are still being researched and await clinical validation. EEG as a promising candidate remains obtrusive, costly, time-consuming, and hard to apply in real-life situations. ECG, even though more common in wearable devices, lacks specificity in its correlation to sleepiness and/or fatigue and sensitivity to lower degrees of sleep restrictions [22]. Body temperature is also an interesting measure as it was previously linked to sleep and sleep regulation [23,24,25]. Contrary to core body temperature, skin temperature can be assessed relatively unobtrusively, at a lower cost, and relates to subjective sleepiness and performance decrements [19,26,27,28]. In their sleep deprivation study, Romeijn and colleagues found a significant effect of the sleep condition (normal sleep vs. total sleep deprivation) on skin temperature gradients. Furthermore, they found a significant relation between skin temperature gradients and performance markers (reaction speed and lapses) in each of the sleep conditions separately. Yet, their results highlighted no statistical difference in the temperature–performance relation between sleep conditions. Further investigation of the link between skin temperature and the consequences of sleep restriction for daytime performance and experiences is needed to assess the relevance and potential utility of this physiological marker in future studies or clinical applications. Promisingly, a recent study performed in a clinical setting shows a disruption of skin temperature rhythm, associated with the severity of EDS, amongst obstructive sleep apnea patients [29].

In this field study, we aimed to investigate moderations in self-reports of sleepiness and fatigue and skin temperature metrics as a function of the cumulative effect of sleep debt by imposing a sleep restriction over three consecutive nights. In line with previous findings in the literature [30], we anticipate increased levels of self-reported complaints in the restricted sleep condition, along with higher levels of sleepiness and fatigue on the final day compared to the first day of sleep restriction.

The study makes several contributions to the literature. First, in contrast to previous studies, we employed ecological momentary assessment with the aim of capturing real-world, within-day fluctuations in sleepiness and fatigue following disturbed sleep. Previous research has revealed different types of temporal patterns in daytime complaints [6,19,20,21]. As such, we will investigate both linear and curvilinear patterns, along with how sleep conditions may affect these patterns. Second, by systematically comparing multiple subjective scales and examining their convergence, we aim to provide new insights into lived experiences of daytime complaints and how their monitoring in clinical samples might be improved. Based on the previously mentioned reflections on common questionnaires, we expect momentary measures to be more sensitive than daily assessments. Third, we explored the potential of skin temperature metrics as relatively unobtrusive physiological proxies for detecting sleepiness and fatigue. Our main hypothesis, based on previous research [31,32,33], is that a higher distal skin temperature during the prior hour could be linked to higher daytime complaints. In fine, this investigation offers a novel perspective on both the temporal dynamics and physiological correlates of daytime complaints under sleep restriction, with implications for improving clinical assessment and monitoring strategies.

## 2. Results

### 2.1. Adherence to the Ecological Momentary Assessment Protocol

After completion of the study and pre-processing of the data, we counted 266 sleep diary entries, 87 daily assessments (ESS, PROMIS), and 807 momentary assessments (KSS, SSS, VAS fatigue). On average, the response rates of the 17 participants were 92.04% (SD = 9.05%, range = 70.59–100%) for the sleep diaries, 85.29% (SD = 17.56%, range = 50–100%) for the daily assessments of sleepiness and fatigue, and 79.12% (SD = 11.14%, range = 61.67–95%) for the momentary assessments of sleepiness and fatigue.

### 2.2. Sleep Schedule Adherence

Participants followed the sleep schedule instructions (see Appendix A for descriptive statistics of sleep quality, duration, and timings). Compared to the baseline, participants slept on average 2% (SD = 10.8%) more in the NS condition and 49% (SD = 4.7%) less in the RS condition. The average sleep durations were 8.16 h (SD = 0.77) in the NS condition and 4.06 h (SD = 0.34) in the RS condition. A total of 11 participants mentioned dozing off during the day (ranging from once to four times a day) in different phases of the study (baseline, NS, RS, washout period). The average total duration of dozing events in minutes during the RS condition was relatively short (M = 15.9, median = 7.5, SD = 16.8).

### 2.3. Impact of (Cumulative) Sleep Restriction on Self-Reports and Thermophysiology

#### 2.3.1. Effect of Sleep Restriction on Daily Self-Reports

Figure 1 represents the effect of sleep restriction with higher daily scores of sleepiness (2a) and fatigue (2b) in orange as compared to the scores in the NS condition (in blue).

The effect of the sleep condition on the ESS score was significant with, on average, a higher ESS score in the RS condition (estimated marginal mean ± standard error = 11 ± 1.13) as compared to the NS condition (4.2 ± 0.77) (see Table 1 for the test statistics). We observed a significant main effect of Day on the ESS, suggesting an increase in the ESS score from Day 1 (6.54 ± 0.86) to Day 2 (7.6 ± 0.87) and to the highest value on Day 3 (8.61 ± 0.89).

Results on daily fatigue (PROMIS) mirrored the ones of the ESS, with a significant main effect of Condition, marking a higher level of fatigue in the RS condition (13.31 ± 1.10) than in the NS condition (5.74 ± 0.81), and a main effect of Day (Day 1 = 8.45 ± 0.82; Day 2 = 9.03 ± 0.84; Day 3 = 11.11 ± 0.87) (Figure 1b). The interaction term Condition*Day did not significantly affect daily fatigue (*p* = 0.31).

#### 2.3.2. Effects of Sleep Restriction on Momentary Measures and Skin Temperature Metrics

The main effect of the sleep condition on momentary sleepiness (KSS and SSS) and fatigue (VAS_fatigue_) was not statistically significant (see Table 2 for the test statistics). SSS and VAS_fatigue_ scores significantly increased over the days (main effect of the variable Day), with values rising each day (SSS: Day 1 = 2.81 ± 0.13; Day 2 = 2.9 ± 0.13; Day 3 = 3.25 ± 0.13; VAS_fatigue_: Day 1 = 3.72 ± 0.31; Day 2 = 3.83 ± 0.31; Day 3 = 4.17 ± 0.32). This main effect of Day was not statistically significant for KSS (Day 1 = 3.62 ± 0.20; Day 2 = 3.78 ± 0.20; Day 3 = 3.95 ± 0.20). As can be observed in Figure 2 comparing scores in normal sleep (in blue) vs. restricted sleep (in orange), we observed a Condition*Day interaction effect on KSS (Figure 2a) and VAS_fatigue_ (Figure 2c) scores, reflecting a cumulative effect of the sleep restriction over the days. While Figure 2b might suggest it, this cumulative effect was not statistically significant for SSS scores (*p*-value higher than the set alpha criterion of 0.01; see Figure 2b).

Post hoc analyses on the interaction between Condition and Day for KSS and VAS_fatigue_ indicated significant contrasts between the sleep conditions on each of the three experimental days (all *p* < 0.01) (see Appendix A for the estimated marginal means and standard errors, as also depicted in Figure 2). When comparing the assessment days per condition for KSS and VAS_fatigue_, only the contrasts in the restricted sleep condition between Day 1 and Day 3 were significant (*p* < 0.001 for these contrasts). An overview of the post hoc analysis statistics (including those for the SSS) can be found in the Appendix A.

All momentary assessments were significantly influenced by time of day (Time) (KSS: B = −0.89, SE = 0.15, β = −2.21, CI = [−2.96, −1.47]; SSS: B = −0.49, SE = 0.12, β = −1.66, CI = [−2.42, −0.90]; VAS_fatigue_: B = −0.81, SE = 0.18, β = −1.42, CI = [−2.04, −0.80]) and by Time-of-day squared (Time_sqr) (KSS: B = 0.03, SE< 0.01, β = 2.25, CI = [1.51, 2.98]; SSS: B = 0.02, SE< 0.01, β = 1.82, CI = [1.07, 2.58]; VAS_fatigue_: B = 0.03, SE = 0.01, β = 1.61, CI = [1.00, 2.22]), indicating that time-of-day variations in subjective sleepiness and fatigue follow, on average across all observations, a curvilinear function (with lower levels at midday compared to morning and evening, represented by the green lines in Figure 3) rather than a linear function. The interaction terms between Time or Time_sqr and Condition were not statistically significant (see Table 2), suggesting that the time-of-day variations in subjective scores occurred independent of the sleep condition in the current sample.

The proportion of variance explained by fixed effects (R2 fixed effect) appears to be of similar size between the momentary measures of sleepiness (KSS or SSS) and fatigue (VAS_fatigue_) (ranging from 18 to 20%; see Table 2). The random slope for Condition was significant for the model on KSS (variance = 0.75, *p* < 0.001) and SSS (variance = 0.30, *p* < 0.001), indicating interindividual differences in how the participants responded to the sleep restriction manipulation. The visual representation of time-of-day variations in momentary assessments shows differences in participants’ patterns of sleepiness and fatigue throughout the day, as represented by individual grey lines in Figure 3. Although visual inspection suggests interindividual differences, incorporating random slopes to account for these variations did not significantly enhance model fit and was therefore disregarded.

The sleep condition significantly affected DST and PST (main effect of Condition), with a higher skin temperature in the RS condition (DST = 30.8 ± 0.27; PST = 34.7 ± 0.13) as compared to the NS condition (DST = 30.4 ± 0.27; PST = 34.5 ± 0.13). Skin temperature (DST, PST) and temperature gradient (DPG) were not significantly affected by Day or by the Condition*Day interaction effect (see Table 2). Factors relating to time of day (Time, Time_sqr) significantly affected both PST and DPG, highlighting a time-of-day-specific curvilinear variation of the temperature. This pattern was affected by the sleep condition for PST, as indicated by the statistically significant interaction terms for Condition*Time and Condition*Time_sqr. These interaction terms were not statistically significant for DPG. Ambient temperature did not significantly improve the linear mixed models testing the impact of sleep restriction (likelihood ratio test comparing models with and without interaction term with ambient temperature, *p* > 0.01).

### 2.4. Convergence Between Subjective Sleepiness and Fatigue and Skin Temperature Metrics

#### 2.4.1. Convergence Between Subjective Measures of Sleepiness and Fatigue

Daily fatigue significantly related to daily assessments of sleepiness (ESS) (F1/70 = 209.36, *p* < 0.001, B = 0.76, SE = 0.05). The correlational strength was found to be medium to large, as represented by a β of 0.79 (CI = [0.68, 0.89]).

KSS scores were significantly related to assessments made by the SSS and the VAS_fatigue_. Similarly, SSS scores significantly related to the VAS_fatigue_ scores. In all combinations, standardized parameters were positive and suggested moderate-to-high correlations between the momentary self-reports. The standardized coefficients appeared to have a rather similar order of magnitude for the three models (see Table 3).

Relations between the daily sleepiness measure (ESS) and the daily aggregated features of KSS, SSS, and VAS_fatigue_ were significant, except for the first KSS of the day, which did not significantly relate to daily ESS. The correlational strengths in significant relations were typically moderate, with the highest parameter estimate (β) belonging to the mean of each momentary assessment (see Table 4).

The relations between momentary assessment of sleepiness and fatigue (KSS, SSS, VAS_fatigue_) and daily measures of fatigue (PROMIS) mimicked the relations with daily measures of sleepiness (ESS) (see Appendix A).

#### 2.4.2. Relation Between Momentary Subjective Sleepiness or Fatigue and Preceding Hourly Skin Temperature

DST and DPG significantly related to KSS, SSS, and VAS_fatigue_ (see Table 5). A higher DST in the prior hour coincided with a higher level of sleepiness and fatigue at the end of the hour (as represented by the black lines in Figure 4a,d,g). DPG behaved the opposite, with a higher gradient related to lower sleepiness and fatigue (Figure 4c,f,i). Proximal skin temperature was significantly related to only the SSS measure, showing a positive relation (see Figure 4b,e,h). The correlational strength in significant models appeared modest.

## 3. Discussion

The current study aimed to offer new insights into the temporal fluctuations in daytime complaints during normal and restricted sleep using ecological momentary assessment. Moreover, we compared the responsiveness of the momentary sleepiness and fatigue ratings to sleep restriction with that of daily sleepiness and fatigue assessments. Additionally, we investigated, for the first time, the feasibility and potential of skin temperature metrics as minimally obtrusive physiological proxies for sleepiness and fatigue by examining the convergence between momentary self-reported measures and skin temperature from the preceding hour. Using ecological momentary assessment, we quantified both between- and within-day variations in the effects of sleep restriction on daytime complaints. This allows us to reflect on the responsiveness of common subjective sleepiness assessments (ESS, KSS, and SSS) over time and examine the potential of skin temperature as an unobtrusive and fine-grained proxy for detecting daytime complaints.

As expected from prior literature [30,34,35], restricting sleep significantly increased participants’ perceived sleepiness and fatigue compared to a normal sleep schedule. Our results highlight that the effects of sleep restriction were evident after the first night, which aligns with findings from studies on single-night sleep restriction [36]. Participants’ scores on the Epworth Sleepiness Scale (ESS) quickly surpassed the clinical cutoff for excessive daytime sleepiness (EDS). Interestingly, while the momentary subjective assessments of sleepiness (Karolinska Sleepiness Scale; KSS) and fatigue (Visual Analog Scale for Fatigue; VAS_fatigue_) also indicated an effect of sleep restriction, the reported sleepiness severity was less pronounced than according to the ESS. This was demonstrated by an average rating of “5—Neither alert nor sleepy” on the KSS on the last day of the sleep restriction condition. In clinical practice, the scales used to assess sleepiness, especially the ESS, KSS, and SSS, are generally believed to reflect the severity of daytime sleepiness in a similar way, even if they operate on different time scales. However, our results suggest otherwise. Participants may have subjectively perceived the burden of sleep restriction differently throughout the day, potentially less severe when actively engaged in activities as a compensatory mechanism, compared to later in the day when sleep pressure typically peaks. Alternatively, the scales may assess different dimensions of sleepiness, as previously hypothesized [6,37]. Nevertheless, in our study, the average, maximum, minimum, and last momentary assessments of the day each correlated only moderately with the daily ESS, demonstrating relatively consistent, yet not strong, convergence between assessment tools. Given that the relevance of the ESS is increasingly debated, partly due to low repeatability [38], (clinical) assessments of sleepiness could instead be performed repeatedly with shorter scales capturing momentary states, like the SSS or the KSS. One advantage of repeated momentary scales is their ability to monitor the cumulative effects of chronic sleep restriction. Although the first night of restricted sleep showed its toll on both daily and momentary metrics, only the momentary measures reflected the added burden of subsequent sleep loss. This aligns with the findings of Dinges and collaborators [30], who identified a cumulative toll of sleep restriction, marked by a peak in subjective sleepiness following multiple nights of sleep restriction.

Beyond the cumulative effects, momentary assessments also highlighted within-day variations. Across all observations, subjective daytime sleepiness and fatigue appeared to vary throughout the day in a curvilinear manner (U-shape), with, on average, higher levels during the morning and evening, as compared to during the afternoon. With such observations, we align with other authors in highlighting a circadian force on top of the homeostatic process [22,39,40,41,42,43]. We accounted for individual differences in the severity of daily (ESS, PROMIS) and momentary (KSS, SSS, VAS_fatigue_) daytime complaints by allowing the intercept to vary across participants. Furthermore, random slopes addressed variations in participants’ responses to the restriction. Inspecting random slopes, our statistical models revealed no statistically significant differences in time-of-day patterns of subjective sleepiness and fatigue between participants, likely due to the limited sample size. Yet, other studies have observed different types of patterns across the day: a steady increase in sleepiness over time awake [41] or a sinusoidal variation [44]. The potential differences in daily patterns among participants remind us of the experiences of patients treated for obstructive sleep apnea [6], which points to a clinical relevance in studying interindividual variations in time profiles. Schneider et al. [20] hinted at different daytime variations in tiredness between narcolepsy, insomnia, and obstructive sleep apnea patients. Perhaps these temporal dynamics are related to a person’s vulnerability (or lack thereof) to sleep debt’s effects [40]. More research is needed to investigate the extent to which people affected by repeated sleep debt present (inter)individual differences in the temporal dynamics of daytime complaints and how these time-of-day patterns relate to their severity or diagnosis. To our knowledge, variations in complaints (and stability of this variation between days) are typically not considered in the diagnosis and clinical monitoring of EDS. In fact, the ESS, perhaps the most commonly used metric to determine EDS, cannot—by its nature—reflect any within-day variations in daytime complaints or day-to-day variations when using a typical 2-week look-back period.

Interestingly, assessments of sleepiness versus fatigue showed substantial convergence (both on a momentary and daily basis). Moreover, the effects of the sleep manipulation on both subjective sleepiness and fatigue were of similar magnitudes, indicating that these states blur or coincide. These observations align with findings from patients experiencing daytime complaints related to sleep disorders such as obstructive sleep apnea and narcolepsy [6]. The substantial overlap between the sleepiness and fatigue assessments on a momentary basis (as represented by the rather high standardized parameter estimates) deviates from earlier studies distinguishing the two concepts theoretically and empirically [7,9]. Rather, we match recent findings by Baek and colleagues [45], and we align with the idea of overlapping constructs or that fatigue and sleepiness are interchangeable terms [6], furthermore pointing at the complexity of monitoring daytime complaints. Both current and past results [6] point at the possibility of assessing daytime complaints such as sleepiness differently, perhaps by using various scales and formulations to better address patients’ complaints and provide a more comprehensive monitoring of their lived experiences. Future research should continue exploring daytime complaints in real-world settings and seek clinical validation of momentary scales.

In parallel to our efforts in revisiting common subjective scales to measure sleepiness, it is important to investigate other avenues to improve the assessment of daytime sleepiness, such as through thermic proxies. Unfortunately, the relation between sleep, sleepiness, and skin temperature is not straightforward, and like in the thermophysiological literature, our results show inconsistencies. Kräuchi and colleagues [31,46] suggested independence of core body and skin temperature from homeostatic sleep pressure, while other authors showed an impact of a 1-night sleep restriction on various skin temperature gradients [26]. In our study, we detected no statistically significant effect of sleep restriction on the gradient but did observe an effect on DST and PST. These inconsistent results between skin temperature metrics underline the need to further investigate and understand the impact of sleep restriction on skin temperature. Results on the impact of time-of-day on PST and DPG align with the knowledge that skin temperature has a circadian rhythm [27]. Although our results indicated an impact of sleep restriction on the time profile of PST, interpretation is limited by the non-conformity of the model to statistical assumptions (normality of the residuals). Further research is needed to investigate to what extent sleep restriction impacts the rhythm of skin temperature.

Our findings on the relation between skin temperatures and subjective metrics do align with earlier literature. Some studies showed [31,32,33] a higher distal skin temperature (increased in vasodilation) was related to higher levels of sleepiness as measured by both the KSS and SSS (within-subjects). This observation resembles the model formulated by Kräuchi [31], in that variation of sleepiness might be related primarily to variation of skin temperature (vasodilation) through the circadian regulatory system. However, contrary to what other studies suggest [26,33], an increase in vasodilation (shown by rising DPG, though calculated between hands and torso) correlated with lower self-rated sleepiness. This may stem from the assessment time. As DPG has been associated with sleep onset [33], earlier studies often assessed it during the time surrounding the sleep episode, whereas we assessed it throughout the entire day.

Although some associations between skin temperature and self-reported sleepiness and fatigue reached statistical significance despite the noisiness of the data from field acquisition, it is important to mention that the relational strength was modest. For skin temperature to be considered a viable proxy or detector of changes in sleepiness or fatigue, more studies conducted in real-world settings are necessary, along with improvements in detection methods (such as sensor sensitivity, sampling rate, and size) and data processing. This should begin with clear guidelines for outlier detection in skin temperature data. Nevertheless, a recent study demonstrated a disrupted skin temperature rhythm among OSA patients (a population known to be affected by EDS), with ties to sleepiness severity [29]. Another investigation among insomniacs (among whom EDS is not considered a symptom characteristic of the diagnosis) showed a stability of the core body temperature rhythm over the days as compared to healthy controls [47]. These few observations underline the necessity to explore the impact of sleep restriction on (core and skin) temperature rhythm and the link to daytime complaints. A clear benefit of a low-obtrusive and continuous monitoring method such as this is that it would allow us to measure and model the variation in complaints during the wake phase in the field over prolonged periods of time and at a large scale.

While some authors are endeavoring to establish a “single measure” that would reliably detect or diagnose sleepiness using, for instance, the psychomotor vigilance task [16], we argue that a single snapshot or aggregation across a long time window would fail to encompass inherent within-day variations in sleepiness (and fatigue). We endeavor to step away from an argumentation over the merits or limits of objective versus subjective measures by instead arguing for the importance of accounting for the impact of disrupted sleep on all aspects of functioning. To do so, we advise to account for daytime complaints like sleepiness or fatigue by using multiple scales at multiple times of day in parallel to psychophysiological metrics, instead of one-snapshot measures targeting experiential or behavioral manifestations.

The experience sampling method allowed us to observe daytime complaints of sleep restriction in real-life settings and in a semi-continuous manner. Care was taken to limit participant burden, and as such, the assessment did not cover the entire 20 h waking period in the restricted sleep condition. The inclusion of the period before sleep onset on restricted sleep nights could have given more insights into phases with more extreme sleep pressure, i.e., at moments when melatonin is well at play and the drive to sleep is fought against. Despite the values of investigating impacts of sleep on daytime complaints in everyday life to facilitate translation of the results into daily practice, field studies are, by nature, less controlled. In particular, it is difficult to establish what other environmental and behavioral factors were at play in the relations we analyzed. Survey notifications throughout the day might have had a direct alerting effect, and participants could have chosen different activities in the restricted sleep versus normal sleep condition to face the effects of sleep restriction. In fact, in a recent qualitative study, patients mentioned the use of various methods, including activities, to mitigate daytime complaints like tiredness, sleepiness, and fatigue [6]. Adjunctive monitoring of the type of activities and contextual differences would be beneficial in further investigations. Another consideration is that our sample, although sufficient to detect a cumulative effect of sleep restriction on daily averages of momentary scores according to the a priori power analysis, was relatively small and consisted mainly of students. A larger and more diverse sample would provide a higher statistical power to detect small effect sizes and could provide stronger generalizability across age groups and environments. Related to this, the statistical power to examine, for instance, slope variation among participants was limited. Given the disparity in assessment frequency between the momentary and daily measures, it would be pertinent to acknowledge the corresponding differences in statistical power to detect equivalent effect sizes. However, the ecological momentary assessment protocol allowed us to obtain repeated measures within subjects, permitting linear mixed models and increasing statistical power.

## 4. Materials and Methods

### 4.1. Design and Manipulation

We performed a randomized cross-over study during which ecological momentary assessment was employed to test the effects of a 3-day sleep restriction (versus 3-day normal sleep) on daily and momentary self-reports of sleepiness and fatigue and skin temperature levels in the hour prior to these assessments. The order of sleep conditions was counterbalanced across participants. In the restricted sleep (RS) condition, participants were asked to follow a sleep schedule with 4 h of time in bed per night for three consecutive nights. Adherence to the instructions was verified by phone calls at scheduled sleep onset (around 04:00) and sleep offset (around 08:00) and with sleep logs and actigraphy. Sleep onset and offset for the RS nights were scheduled to fit with participants’ habitual sleep–wake cycles, with the sleep episode occurring in the second half of a normal night. In the normal sleep (NS) condition, participants slept 8 ± 1 h per night for three consecutive nights.

Subjective sleepiness and fatigue and skin temperature were the primary outcome variables. Self-reports were repeatedly acquired at a momentary and daily level during the experimental conditions. This study is part of a larger pre-registered protocol (doi: 10.17605/OSF.IO/R287D) involving additional behavioral measures (psychomotor vigilance task and keyboard interactions) acquired in the field and in-lab psychophysiological measures obtained on Day 3 of both experimental conditions. These additional measures will not be reported here.

The study protocol is in agreement with the Declaration of Helsinki (2013) and approved by the general Ethical Review Board of the Eindhoven University of Technology (ref. ERB2021IEIS40) and the Human-Technology Interaction Ethical Review Board (ref. 1436). Furthermore, the Medical Ethics Committee of Máxima Medical Center assessed the study as not subject to the Medical Research Involving Human Subjects Act (WMO), i.e., not required to undergo a review by an accredited MREC or the CCMO (ref. N21.075).

### 4.2. Participants

Seventeen healthy participants (19–32 years old, mean = 22.2, SD = 3.73, range = 19–32, 7 females, 10 males) were recruited via a university participant database. All participants were screened for non-extreme chronotype using the Ultra-short Munich Chronotype Questionnaire [48] with age-matched thresholds [49] and no severe disturbances in general sleep quality (Pittsburgh Sleep Quality Index > 5) [50]. An overview of the study’s demographic information is provided in Appendix A. Additional exclusion criteria were: intercontinental travel in the 3 months prior to the study, visual disability or a medical diagnosis, pregnancy, drug use, use of a car to travel to the study facility, and use of heavy machinery for the duration of the study. Participants were asked to limit their daily caffeine consumption (<4 units/day) and their weekly alcohol consumption (14 units/week), keeping to health recommendations. Participants gave written informed consent and were financially compensated for their participation.

Sample size was estimated based on a priori power analysis conducted based on subjective ratings obtained in other studies with a similar paradigm. The details of the power analysis can be found in the pre-registration (doi: 10.17605/OSF.IO/R287D).

### 4.3. Measures

#### 4.3.1. Self-Reports

Sleep timing, duration, and quality were assessed through the Core Consensus Sleep Diary—E version [51]. In the morning half of the sleep diary, participants reported on their prior sleep. In the evening half of the diary, participants reflected upon behaviors during the day (naps, alcohol and caffeine consumptions, sleep medication, etc.).

Momentary subjective sleepiness was measured through the Karolinska Sleepiness Scale (KSS) [52] and Stanford Sleepiness Scale (SSS) [53]. Daily subjective sleepiness was assessed with the ESS [54]. The ESS is a scale often used to estimate the level of sleepiness over the prior two weeks. However, the instructions of the scale (“in recent times”) allow researchers to adapt it to a shorter look-back period (in this case, a day). The Cronbach’s alpha was 0.87, demonstrating good internal reliability.

Faced with the lack of validated scales measuring momentary fatigue, a visual analog scale (VAS_fatigue_) was chosen to assess momentary fatigue from 1 (“not fatigued at all”) to 10 (“extremely fatigued”). The daily adapted Patient-Reported Outcomes Measurement Information System (PROMIS^®^) [55] assessed daily levels of fatigue through seven questions (Cronbach’s a = 0.92). Total scores on the PROMIS^®^ will be referred to as daily fatigue in this report.

#### 4.3.2. Sensor-Derived Data

Participants were equipped with the Actiwatch Spectrum Pro (Philips Respironics, Amsterdam, The Netherlands), measuring gross motor activity at the wrist over the entire study period of 17 days, with a sample every 30 s.

Skin temperature was continuously recorded (sample interval of 300 s, with a resolution of 0.0625 °C) during each experimental condition using four wireless sensors (Thermochron iButtons) attached by surgical tape to both hands (dorsal side) and left and right sub-clavicular regions. Proximal (PST) and distal (DST) skin temperatures were obtained by averaging the values recorded respectively by the sub-clavicular and the hand sensors. The distal-to-proximal skin temperature gradient (DPG) was calculated by subtracting the proximal temperature from the distal temperature [23]. Although the computation of DPG occasionally incorporates skin temperature data collected at more body positions than selected in this study, we have opted to concentrate on monitoring methods conducive to field assessments and imposing a lesser burden on participants. Ambient temperature was recorded through a fifth wireless sensor fixed to a portable clip placed on top of participants’ clothes.

As a first outlier detection and removal step: sudden changes of more than 2 °C in skin temperature were excluded from the data. Subsequent rules of outlier exclusion included: not wearing the sensors (difference between ambient temperature and clavicular temperature below 2 °C), left and right sub-clavicular temperatures differed (>1.5 °C), improbable PST (PST < 31.5 °C or PST > 37 °C), abnormal DST (<20 °C when ambient temperature >17 °C, and DST < ambient temperature in winter and autumn), and a high DPG (absolute value > 10 °C). This pre-processing resulted in the exclusion of 14.8% of the data. Skin temperature data were then mapped with the completion timings of the momentary assessments to compute hourly averages of skin temperature before self-reports of sleepiness and fatigue.

### 4.4. Procedure

The experiment was conducted in Eindhoven, the Netherlands, between January 2022 and May 2023. The study protocol consisted of one baseline week followed by two experimental conditions: normal sleep (NS) and restricted sleep (RS) (Figure 5).

During the baseline week, participants were instructed to maintain a regular sleep schedule of approximately 8 ± 1 h per night for seven consecutive days. Throughout this period, daily sleep diaries were administered via the MetricWire smartphone application (MetricWire Inc., Waterloo, ON, Canada) in the morning and afternoon and actigraphy data was continuously collected. Moreover, smartphone keyboard interactions were monitored during typing episodes to capture dynamics in typing behavior. Sleep diaries, actigraphy, and keyboard monitoring continued throughout the entire study (17 days in total). In addition, participants wore iButtons continuously to measure skin temperature during the three sampling days in each experimental condition.

During the experimental conditions, participants completed experience sampling questionnaires via the MetricWire smartphone application (MetricWire Inc.). Each day, between 08:00 and 23:00, they received ten semi-random notifications, spaced at least 30 min apart, to report on their level of sleepiness and fatigue (KSS, SSS, VAS-fatigue). The timing of the notifications was semi-randomized for each participant, with an equal number distributed across the morning and afternoon/evening.

Participants also received a fixed notification for the psychomotor vigilance task (PVT) at 15:30 and for evening assessments of sleepiness (ESS) and fatigue (PROMIS fatigue) at 22:00. At the end of each condition, the sensors were collected and data were downloaded. After completing the study, participants were thanked and debriefed.

### 4.5. Statistical Analysis

Data processing, visualizations, and statistical analyses were performed in Python (version 3.12.4) and R (version 4.1.2) with the “lme4”, “lmer”, “emmeans”, “seaborn”, “matplotlib”, “pandas”, and “numpy” packages. The alpha criterion was defined at 0.01 to account for the number of models performed per set of variables.

Multilevel analyses using linear mixed models were applied to investigate the cumulative sleep restriction effect on self-reports and skin temperature. Models were run separately for each outcome (KSS, SSS, VAS, ESS, PROMIS, DST, PST, and DPG) and were built iteratively (upon improvement of model fit assessed by likelihood ratio test). Models were built iteratively to ensure optimal model construction and fit. This stepwise approach allows for the systematic inclusion of relevant predictors and random effects while balancing model complexity and interpretability.

The statistical analysis started with the consideration of random intercepts in unconditional models (see Appendix A for the intraclass correlations derived from these models). ParticipantID was always added as a random intercept, sometimes accompanied by Session within ParticipantID if fitting (accounting any potential random effect of order on intraindividual differences). Condition, Day, and the interaction Condition*Day were subsequently added as fixed factors to the unconditional model. Post hoc contrast analyses of the estimated marginal means were performed with Tukey correction (allowing for pairwise comparisons) to further inspect the interaction effect of Condition*Day when significant. To investigate time-of-day dependency in momentary assessments, time-of-day (Time) and time-of-day squared (Time_sqr) were added as well as the interaction terms Condition*Time and Condition*Time_sqr.

Random slopes to account for individual differences in the response trend were added conditional upon the model’s fit improvement (likelihood ratio test, *p* < 0.01). Residuals of the linear mixed models were assessed for normality using the Shapiro–Wilk test (W > 0.97 was used as the cutoff value).

The convergences of outcome variables were also examined using linear mixed models. The model assessing the relationship between daily assessments (ESS and daily fatigue) featured the ESS score as the dependent variable and the person-mean-centered PROMIS score as the predictor variable. The same approach was used to explore convergence among the momentary assessments. In these linear mixed models, KSS was included as the dependent variable and the SSS or VAS_fatigue_ as the predictor variable, while SSS was treated as the dependent variable with respect to VAS_fatigue_. The subjective measures were regarded as dependent variables (in separate models) with DST, PST, or DPG as a predictor. The relationships between daily assessments and various aggregates of momentary assessments per day (i.e., the mean, minimum, maximum, first, and last entries of the day) were also analyzed using linear mixed models. ESS and PROMIS were included as the dependent variable, while the aggregated scores of the momentary assessment were incorporated as fixed predictors. In these models, ParticipantID was added as a random intercept and the predictor variables were person-mean-centered. Standardized parameters were derived through the standardize_parameters function of the Parameters package in R, utilizing the “pseudo” method and the Satterthwaite method for estimating confidence intervals.

## 5. Conclusions

Distal and proximal skin temperature appeared to be affected by sleep restriction, although the temperature gradient was not. Both distal skin temperature and the temperature gradient showed relations to momentary self-reports of sleepiness and fatigue. While daily assessments highlighted the general effects of sleep manipulation, momentary assessments uniquely demonstrated a cumulative impact of ongoing sleep restriction over several days. They revealed variations within the same day, independent of sleep duration—something daily assessments cannot address. Our results also show that the severity of sleepiness caused by heavy sleep restriction is not consistently represented across different scales and that the time of day can significantly influence this assessment of complaints. Significant individual differences in responses underscore the need for personalized approaches when addressing daytime issues, such as sleepiness and fatigue. Therefore, we recommend that clinicians incorporate repeated momentary scales when evaluating patients’ daytime complaints. This addition could improve diagnosis and help tailor treatment to individual patterns of daytime symptoms.

## Figures and Tables

**Figure 1 clockssleep-07-00051-f001:**
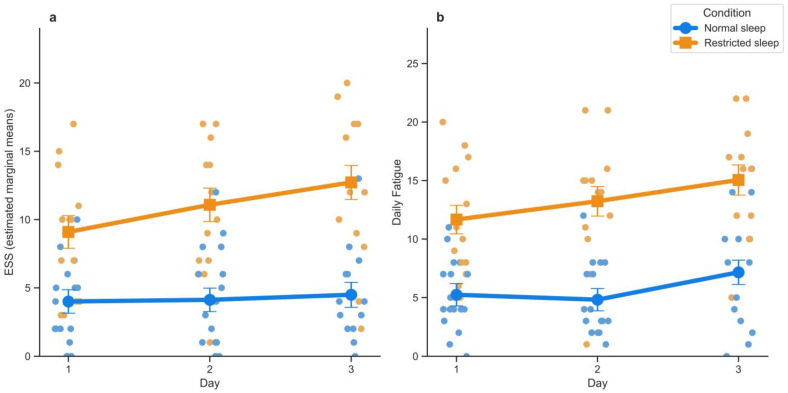
Effects of sleep condition (Normal sleep: blue line with dots, Restricted sleep: orange line with squares) and of experimental days (Day 1 to Day 3) on daily measured sleepiness (ESS, (**a**)) and fatigue (PROMIS, (**b**)), respectively. Values correspond to estimated marginal means, error bars show standard errors, and scatter represents all entries of the scale per assessment (Restricted sleep: blue; Normal sleep: orange).

**Figure 2 clockssleep-07-00051-f002:**
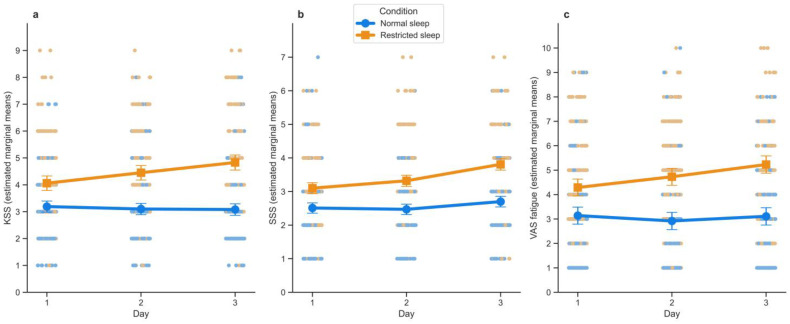
Estimated marginal means of sleepiness and fatigue over the experimental days in each sleep condition. Effects of sleep condition (Normal sleep: blue line with dots, Restricted sleep: orange line with squares) and experimental days (Day 1 to Day 3) on momentary sleepiness and fatigue levels, measured through the Karolinska Sleepiness Scale (KSS, (**a**)), Stanford Sleepiness Scale (SSS, (**b**)), and fatigue (VAS_fatigue_, (**c**)) sampled up to ten times per day. Values correspond to estimated marginal means over the days, error bars show standard errors, and scatter represents all responses to the corresponding scale (Restricted sleep: orange; Normal sleep: blue).

**Figure 3 clockssleep-07-00051-f003:**
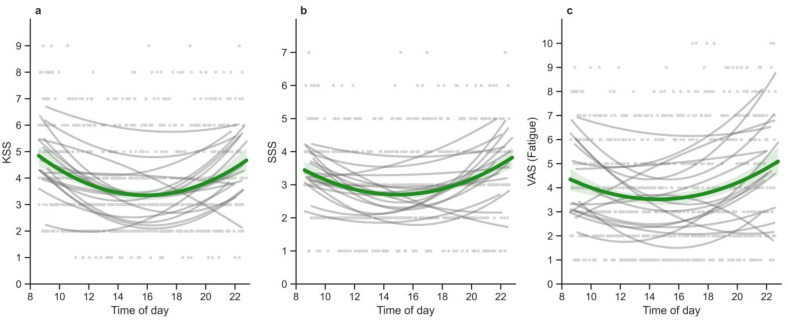
Momentary sleepiness (KSS (**a**), SSS (**b**), and fatigue (VAS_fatigue_ (**c**)) over the 24 h day (independent of the sleep condition). The green lines represent the average subjective rating over time of day across all participants. Grey scatter represents all entries of the scale, while each grey line represents the data of one participant.

**Figure 4 clockssleep-07-00051-f004:**
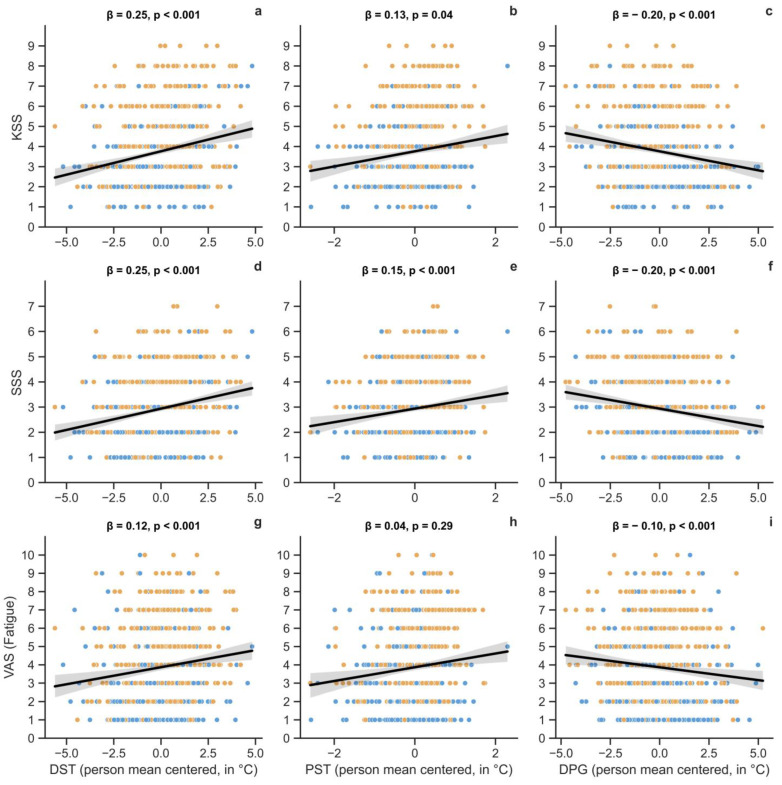
Momentary sleepiness (KSS, SSS) and fatigue (VAS) in relation to skin temperature (hourly aggregate before self-reports prompts): KSS-DST (**a**), KSS-PST (**b**), KSS-DPG (**c**), SSS-DST (**d**), SSS-PST (**e**), SSS-DPG (**f**), VAS-DST (**g**), VAS-PST (**h**), VAS-DPG (**i**). DST = Distal skin temperature, PST = Proximal skin temperature, DPG = distal-to-proximal skin temperature gradient, β = standardized parameter (for person-mean-centered temperatures), *p* = *p*-value. Scatters represent the data entries in the restricted sleep condition (orange) and normal sleep condition (blue).

**Figure 5 clockssleep-07-00051-f005:**
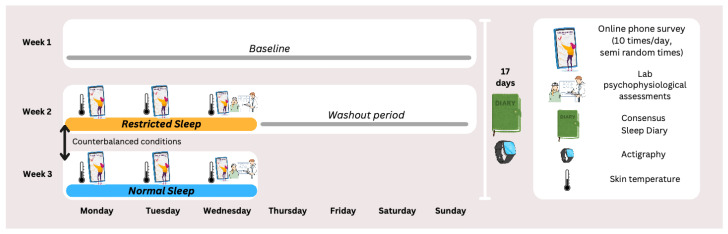
Overview of the procedure and assessment tools included in this investigation.

**Table 1 clockssleep-07-00051-t001:** Model statistics for both ESS and daily fatigue (PROMIS). Both models included ParticipantID as a random intercept, with random slopes accounting for individual differences in response to Condition.

	Condition		Day		Condition*Day		R^2^ Fixed Effect	R^2^ Full Model	AIC
	F	*p*	F	*p*	F	*p*			
ESS	**F_(1/17)_ = 44.67**	**<0.001**	**F_(2/55)_ = 7.72**	**<0.01**	F_(2/55)_ = 4.62	0.01	0.42	0.88	449.72
Daily fatigue	**F_(1/17)_ = 39.06**	**<0.001**	**F_(2/56)_ = 7.91**	**<0.001**	F_(2/56)_ = 1.19	0.31	0.47	0.82	483.34

F = F statistics, *p* = *p*-value, AIC = Akaike information criterion. R^2^ fixed effect: variance explained by main and interaction effects, R^2^ full model: variance explained by fixed and random effects. Statistically significant effects (*p* < 0.01) are presented in bold.

**Table 2 clockssleep-07-00051-t002:** Model statistics for momentary assessments (KSS, SSS, VAS_fatigue_) and skin temperature metrics (DST, PST, DPG).

	Condition		Day		TOD		TOD^2^		Condition*Day		Condition*TOD		Condition*TOD^2^		R^2^ Fixed Effect	R^2^ Full Model	AIC
	F	*p*	F	*p*	F	*p*	F	*p*	F	*p*	F	*p*	F	*p*			
KSS	F_1/792_ = 0.85	0.36	F_2/779_ = 3.77	0.02	**F_1/777_ = 71.28**	**<0.001**	**F_1/777_ = 70.93**	**<0.001**	**F_2/779_ = 6.88**	**<0.01**	F_1/778_ = 0.01	0.94	F_1/778_ = 0.05	0.82	0.20	0.43	2879.06
SSS	F_1/792_ = 1.99	0.16	**F_2/779_ = 13.18**	**<0.001**	**F_1/778_ = 45.40**	**<0.001**	**F_1/777_ = 51.36**	**<0.001**	F_2/780_ = 4.02	0.02	F_1/778_ = 0.26	0.61	F_1/778_ = 0.11	0.74	0.18	0.36	2418.01
VAS_fatigue_	F_1/791_ <0.01	0.99	**F_2/778_ = 5.36**	**<0.01**	**F_1/777_ = 27.82**	**<0.001**	**F_1/776_ = 34.33**	**<0.001**	**F_2/779_ = 6.23**	**<0.01**	F_1/777_ = 1.56	0.21	F_1/777_ = 2.37	0.12	0.19	0.50	3158.32
PST **	**F_1/681_ = 7.13**	**<0.01**	F_2/670_ = 4.47	0.01	**F_1/670_ = 34.33**	**<0.001**	**F_1/670_ = 29.20**	**<0.001**	F_2/670_ = 3.48	0.03	**F_1/670_ = 8.38**	**<0.01**	**F_1/671_ = 8.18**	**<0.01**	0.07	0.41	1388.18
DPG	F_1/679_ = 0.01	0.94	F_2/671_ = 2.28	0.10	**F_1/671_ = 20.06**	**<0.001**	**F_1/670_ = 16.51**	**<0.001**	F_2/671_ = 2.87	0.06	F_1/671_ = 0.03	0.87	F_1/672_ = 0.02	0.88	0.04	0.38	2722.76
DST *	**F_1/687_ = 8.87**	**<0.01**	F_2/682_ = 4.91	0.01					F_2/682_ = 3.81	0.02					0.03	0.29	2773.32

F = F statistics, *p* = *p*-value, KSS = Karolinska Sleepiness Scale, SSS = Stanford Sleepiness Scale, VAS = visual analog scale of fatigue, DST = distal skin temperature, PST = proximal skin temperature, DPG = distal-to-proximal skin temperature gradient, AIC = Akaike information criterion, R^2^ fixed effect: variance explained by main and interaction effects, R^2^ full model: variance explained by fixed and random effects. The model for DST included ParticipantID as a random intercept. Models for KSS and SSS included ParticipantID as random intercept with random slopes accounting for individual differences in response to Condition. The VAS_fatigue_, PST, and DPG models included ParticipantID and Session within ParticipantID as random intercepts. * The addition of the factors TOD, TOD^2^, and the interaction terms Condition*TOD and Condition*TOD^2^ did not improve fit and were thus disregarded from the model. ** Residuals of the model were not deemed normally distributed; caution is necessary to interpret the results. Statistically significant effects (*p* < 0.01) are presented in bold.

**Table 3 clockssleep-07-00051-t003:** Summary of statistics on the relation between momentary sleepiness (KSS or SSS) and momentary fatigue (VAS_fatigue_).

DV	IV	F	*p*	B	SE	β	CI
KSS	SSS	F_1/790_ = 1767.3	**<0.001**	1.13	0.03	0.82	[0.78, 0.86]
	VAS *	F_1/17_ = 126.28	**<0.001**	0.60	0.05	0.71	[0.58, 0.84]
SSS	VAS *	F_1/16_ = 143.09	**<0.001**	0.47	0.04	0.75	[0.61, 0.88]

DV = Dependent variable; IV = Independent variable. F = F statistics, *p* = *p*-value, KSS = Karolinska Sleepiness Scale, SSS = Stanford sleepiness scale, VAS = visual analog scale of fatigue, B: parameter estimate, SE = standard error, β = standardized parameter, CI = confidence interval. Independent variables were person-mean-centered. All models included ParticipantID as a random intercept. * Random slope were added as it improved model fit. Random slope accounted for individual differences in the effect of VAS_fatigue_ (as IV). Statistically significant effects (*p* < 0.01) are presented in bold.

**Table 4 clockssleep-07-00051-t004:** Summary of statistics on the relation between daily aggregates of momentary sleepiness (KSS or SSS) or fatigue (VAS_fatigue_) and daily sleepiness (ESS).

		F	*p*	B	SE	β	CI
KSS	Mean	**F_1/70_ = 82.33**	**<0.001**	3.29	0.36	0.67	[0.53, 0.82]
	Min.	**F_1/77_ = 24.61**	**<0.001**	2.46	0.50	0.50	[0.30, 0.70]
	Max. *	**F_1/19_ = 30.09**	**<0.001**	1.78	0.32	0.60	[0.37, 0.83]
	First	F_1/81_ = 5.66	0.02	0.70	0.29	0.27	[0.04, 0.49]
	Last	**F_1/82_ = 27.9**	**<0.001**	1.46	0.28	0.55	[0.34, 0.76]
SSS	Mean	**F_1/71_ = 79.36**	**<0.001**	4.73	0.53	0.67	[0.52, 0.82]
	Min.	**F_1/78_ = 24.56**	**<0.001**	3.50	0.71	0.51	[0.31, 0.72]
	Max.	**F_1/78_ = 49.28**	**<0.001**	2.39	0.34	0.63	[0.45, 0.81]
	First	**F_1/80_ = 15.38**	**<0.001**	1.61	0.41	0.42	[0.21, 0.64]
	Last	**F_1/83_ = 16.28**	**<0.001**	1.43	0.36	0.43	[0.22, 0.65]
VAS_fatigue_	Mean	**F_1/71_ = 50.22**	**<0.001**	2.29	0.32	0.59	[0.42, 0.76]
	Min.	**F_1/83_ = 8.00**	**<0.01**	1.27	0.45	0.33	[0.10, 0.56]
	Max.	**F_1/78_ = 39.23**	**<0.001**	1.43	0.23	0.58	[0.40, 0.77]
	First **	**F_1/78_ = 23.46**	**<0.001**	1.21	0.25	0.49	[0.29, 0.69]
	Last	**F_1/84_ = 12.30**	**<0.001**	0.85	0.24	0.38	[0.17, 0.60]

F = F statistics, *p* = *p*-value, KSS = Karolinska Sleepiness Scale, SSS = Stanford Sleepiness Scale, VAS = visual analog scale of fatigue, B = parameter estimate, SE = standard error, β = standardized parameter, CI = confidence interval. Models included ParticipantID as a random intercept and predictor variables are person-mean-centered. Statistically significant associations (*p* < 0.01) are presented in bold. * Random slope was added as it improved model fit. ** Residuals of the models performed on the normal sleep data were not deemed normally distributed; caution is necessary to interpret the results.

**Table 5 clockssleep-07-00051-t005:** Overview of statistics on the relation between momentary sleepiness (KSS, SSS) and fatigue (VAS_fatigue_) and skin temperature metrics (hourly aggregate of DST, PST, and DPG before self-reports prompts; person-mean-centered).

		F	*p*	B	SE	β	CI
KSS **	DST	**F_1/681_ = 46.27**	**<0.001**	0.23	0.03	0.25	[0.18,0.32]
	PST *	F_1/16_ = 5.14	0.04	0.33	0.14	0.13	[0.01, 0.26]
	DPG	**F_1/681_ = 28.04**	**<0.001**	−0.19	0.04	−0.20	[−0.27, −0.12]
SSS	DST	**F_1/681_ = 46.72**	**<0.001**	0.17	0.02	0.25	[0.18, 0.32]
	PST	**F_1/681_ = 16.31**	**<0.001**	0.27	0.07	0.15	[0.08, 0.22]
	DPG	**F_1/681_ = 28.64**	**<0.001**	−0.14	0.03	−0.20	[−0.27, −0.13]
VAS_fatigue_	DST	**F_1/669_ = 17.32**	**<0.001**	0.15	0.04	0.12	[0.06, 0.17]
	PST *	F_1/22_ = 1.18	0.29	0.12	0.14	0.04	[−0.04, 0.13]
	DPG	**F_1/670_ = 11.94**	**<0.001**	−0.13	0.04	−0.10	[−0.15, −0.04]

F = F statistics, *p* = *p*-value, KSS = Karolinska Sleepiness Scale, SSS = Stanford Sleepiness Scale, VAS = visual analog scale of fatigue, DST = distal skin temperature, PST = proximal skin temperature, DPG = distal-to-proximal skin temperature gradient, B: parameter estimate, SE = standard error, β = standardized parameter, CI = confidence interval. Models included ParticipantID as a random intercept, and models on VAS_fatigue_ also included Session within ParticipantID as a random intercept. Statistically significant associations (*p* < 0.01) are presented in bold. * Indicates adding a random slope as improving the model’s fit. ** Residuals of the models performed on the sleepiness data were not normally distributed (Shapiro–Wilk test: 0.96 < W < 0.97); caution is necessary to interpret the results.

## Data Availability

The data of this study will be made available upon publication in the Open Science Framework repository under the study registration (doi: 10.17605/OSF.IO/R287D).

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
