# Peer review of "Sleepiness and Fatigue as Consequences of Cumulative Sleep Restriction: Insights from Fine-Grained Subjective Measures and Skin Temperature in the Field"

_2624-5175, 2025, doi:10.3390/clockssleep7030051_

Round 1
Reviewer 1 Report
Comments and Suggestions for Authors
The objective of this article is to evaluate common monitoring tools and explore new perspectives by investigating the response of daily and momentary subjective measures of sleepiness and fatigue, and skin temperature, to sleep restriction, evaluating variations between and during the day and their convergence.
I consider it relevant in this article to indicate that skin temperature is significantly related to subjective sleepiness and fatigue.
I found the sample evaluated unsatisfactory, including only women, and the text does not describe the methodology intuitively.
A patient sample containing only women could bias the research; male patients should be included for better results and interesting conclusions within sleep science and diagnostics.
The groups evaluated are not described in a way that is easy to read for researchers and others interested in this article.
The conclusions are not very clear to me.
I consider the bibliographic references reasonable. Bibliographic references in an article can always be improved.
I believe that articles should be sufficiently transparent to avoid confusion or harm to patients when data and results are extrapolated to humans. I really like it when tables and graphs show statistical analyses accompanied by percentage calculations, making it easier for the reader to read and understand.
Author Response
August 29th, 2025
Subject: Letter to Reviewer 1,
Dear reviewer,
We thank you for your comments and your reviews. We have made changes to the method and the discussion that we hope will answer your concerns. Below we copied your comments in red and respond to them in detail in black.
I found the sample evaluated unsatisfactory, including only women, and the text does not describe the methodology intuitively. A patient sample containing only women could bias the research; male patients should be included for better results and interesting conclusions within sleep science and diagnostics. The groups evaluated are not described in a way that is easy to read for researchers and others interested in this article.
We would like to emphasize that we did not include only female participants. We now clearly state that the sample includes only healthy participants, with both females and males. We have clarified both in the abstract and in the method section.
The conclusions are not very clear to me.
We have adjusted and simplified the conclusion.
I consider the bibliographic references reasonable. Bibliographic references in an article can always be improved.
A few recent and relevant references were added to the manuscript. The bibliography was adapted and verified.
I believe that articles should be sufficiently transparent to avoid confusion or harm to patients when data and results are extrapolated to humans. I really like it when tables and graphs show statistical analyses accompanied by percentage.
We have made changes to the method section as well as the rest of the manuscript to clarify our message. Regarding our figures, we agree, and we ensured that they either represent estimated marginal means, or the appropriate statistical estimates (Figure 5), so that reader can observe the results of our statistical tests. While tables do not include estimated marginal means, we reflect on those in text.
Yours sincerely,
On behalf of the authors,
Vaida Verhoef

Reviewer 2 Report
Comments and Suggestions for Authors
This paper has investigated the response of daily and momentary subjective measures of sleepiness and fatigue, and skin temperature to sleep restriction, assessed between- and within-day variations, and studied their convergence. The work is meaningful. However, some issues should be addressed below.
1. Introduction, the previous work should be discussed with the literature, the problem should be addressed clearly. Now the literature is not enough in this section, please add more.
2. Study objectives and hypotheses in introduction, is it a subsection or another section in this paper, it is not appropriate to place it here, so please revise it.
3. Section 4 may be should be section 2, please revise it.
4. The figures are too bad that can not read clearly, please revise the quality of the figures.
5. So many tables here, so is it possible replace them with the figures?
6. Conclusions should be simplified with the highlights.
7. What is the highlights in this paper, please address them clearly in abstracts, introduction and conclusions.
8. How did the results in this paper applied in actual applications, please add them in the paper.
Author Response
August 29th, 2025
Subject: Letter to Reviewer 2,
Dear reviewer,
We thank you for your detailed comments and your reviews. We have made changes to the manuscript that we hope will answer your concerns. Below we copied your comments in red and respond to them in detail in black.
- Introduction, the previous work should be discussed with the literature, the problem should be addressed clearly. Now the literature is not enough in this section, please add more.
We included additional details from the literature specifically on the relation skin temperature and vigilance. (lines 97-107).
- Study objectives and hypotheses in introduction, is it a subsection or another section in this paper, it is not appropriate to place it here, so please revise it.
We now incorporated the subsection in the introduction.
- Section 4 may be should be section 2, please revise it.
Sections have been moved, and the numbering of figures and supplementary tables has been adapted.
- The figures are too bad that can not read clearly, please revise the quality of the figures.
The contrast between colors in the figures has been increased, as well as the overall quality (dpi) of the figures.
- So many tables here, so is it possible replace them with the figures?
Where possible, we use figures instead of tables. However, we chose to present complex analyses involving multiple variables through tables, as they allow the reader to see the statistical estimates and significance levels more clearly.
- Conclusions should be simplified with the highlights.
We have adjusted the highlights and simplified the conclusion.
- What is the highlights in this paper, please address them clearly in abstracts, introduction and conclusions.
We have updated the abstract, introduction, and conclusions to highlight our study's main findings more clearly.
- How did the results in this paper applied in actual applications, please add them in the paper.
In our discussion, we consider how this study paves the way for future research and some methodological improvements on several occasions. However, we also note that additional studies conducted among patients are necessary before drawing conclusions about their application in a clinical setting, such as in diagnosing and monitoring daytime complaints like sleepiness in sleep-disordered patients.
Yours sincerely,
On behalf of the authors,
Vaida Verhoef

Reviewer 3 Report
Comments and Suggestions for Authors
The manuscript is well-written and informative, introducing skin temperature as a novel metric in sleep research. However, it is lengthy and can be significantly shortened as some points are reiterated multiple times.
Key findings, including results and hypothesis testing (test statistics and p-values), should be included in the abstract.
The section on sensor-derived data preprocessing could benefit from more detail, especially on handling outliers. Please provide a rationale for the chosen sample size and discuss potential limitations related to it.
The results section is dense and could be improved with summary statements at the end of each subsection. Some tables and figures could benefit from additional explanations in the text to enhance readability.
In various tables, the authors reported F statistics as F(df1, df2) = F-value and p-value. While this is appropriate, the rows in some of the tables look unusual. The authors might want to think about ways to improve the presentation.
On line 315, there is an empty space before the period.
The colors in the figures are light, and better contrast would be useful. The green curve in Figure 3 should be explained.
Author Response
August 29th, 2025
Subject: Letter to Reviewer 3,
Dear reviewer,
We appreciate your comments and reviews. We have made changes to the manuscript that we hope will answer your concerns. Below we copied your comments in red and respond to them in detail in black.
The manuscript is well-written and informative, introducing skin temperature as a novel metric in sleep research. However, it is lengthy and can be significantly shortened as some points are reiterated multiple times.
Thank you for the suggestion. We have made changes and revisions throughout the manuscript to reduce redundancies and simplify the wording.
Key findings, including results and hypothesis testing (test statistics and p-values), should be included in the abstract.
We have reformulated the key results in the abstract. In order to respect the journal style and avoid going further above the word limit, we have chosen not to include the statistics in the abstract.
The section on sensor-derived data preprocessing could benefit from more detail, especially on handling outliers. Please provide a rationale for the chosen sample size and discuss potential limitations related to it.
We updated the description of the data preprocessing for the skin temperature data in the revised manuscript. Note that we did perform outlier detection and removed observations labeled as outliers (see lines 211-217).
The rationale for sample size is mentioned in lines 172-174. Potential limitations related to the sample size are reflected upon in the discussion (lines 705-714).
The results section is dense and could be improved with summary statements at the end of each subsection. Some tables and figures could benefit from additional explanations in the text to enhance readability.
Thank you for your suggestions. We have revised the subtext of the tables and figures and reference them in more detail in the results section. Following the editor’s advice and the recommendations of fellow reviewers, we made sure not to lengthen the manuscript or create redundancies.
In various tables, the authors reported F statistics as F(df1, df2) = F-value and p-value. While this is appropriate, the rows in some of the tables look unusual. The authors might want to think about ways to improve the presentation.
Thank you for the suggestion. We have checked the tables and updated the formatting for Table 2.
On line 315, there is an empty space before the period.
Thank you for noticing this. We have removed the empty space.
The colors in the figures are light, and better contrast would be useful. The green curve in Figure 3 should be explained.
The colors in the figures have been adjusted. The green curve represents the average pattern across all participants. This is described in the caption (see Figure 4 in the revised manuscript).
Yours sincerely,
On behalf of the authors,
Vaida Verhoef

Round 2
Reviewer 1 Report
Comments and Suggestions for Authors
I believe the rest of the article is very well described and presented in this text, with a summary, introduction, methodology, tables, graphs showing entropy, and discussion.
I only suggest a small correction to the conclusion.
I recommend that this part of the text go to the discussion and leave the conclusion: "Future research should continue exploring daytime complaints in real-world settings and seek clinical validation of momentary scales. Distal and proximal skin temperature appeared to be affected by sleep restriction, although the temperature gradient was not. Both distal skin temperature and the temperature gradient showed relations to momentary self-reports of sleepiness and fatigue. However, more studies conducted in real-life settings are needed before recommending skin temperature as a method for monitoring daytime complaints... , significant individual differences in responses underscore the need for personalized approaches when addressing daytime issues, such as sleepiness and fatigue, and in studying their relationship with thermophysiology."
Author Response
September 4th, 2025
Subject: Letter to Reviewer 1,
Dear reviewer,
We thank you for your comments and suggestions. We have made the requested changes to the conclusion, focusing only on our results, including those on skin temperature. As you suggested, references to future studies have been moved to the discussion. Changes in the manuscript are highlighted in yellow.
Yours sincerely,
On behalf of the authors,
Vaida Verhoef
Reviewer 2 Report
Comments and Suggestions for Authors
Revised well.
Author Response
September 4th, 2025
Subject: Letter to Reviewer 2,
Dear reviewer,
Thank you for taking the time to review the revision and for your positive feedback.
Yours sincerely,
On behalf of the authors,
Vaida Verhoef